# Empirical Study of the Decision Region and Robustness in Deep Neural Networks

## Abstract

In general, Deep Neural Networks (DNNs) are evaluated by the generalization performance measured on unseen data excluded from the training phase. Along with the development of DNNs, the generalization performance converges to the state-of-the-art and it becomes difficult to evaluate DNNs solely based on this metric. The robustness against adversarial attack has been used as an additional metric to evaluate DNNs by measuring their vulnerability. However, few studies have been performed to analyze the adversarial robustness in terms of the geometry in DNNs. In this work, we perform an empirical study to analyze the internal properties of DNNs that affect model robustness under adversarial attacks. In particular, we propose the novel concept of the *Populated Region Set (PRS)*, where trained samples are populated more frequently, to represent the internal properties of DNNs in a practical setting. From systematic experiments with the proposed concept, we provide empirical evidence to validate that a low PRS ratio has a strong relationship with the adversarial robustness of DNNs.

## 1 Introduction

With the steep improvement of the performance of Deep Neural Networks (DNNs), their applications are expanding to the real world, such as autonomous driving and healthcare (LeCun et al., 2015; Miotto et al., 2018; Huang & Chen, 2020). For real world application, it may be necessary to choose the best model among the candidates. Traditionally, the generalization performance which measures the objective score on the test dataset excluded in the training phase, is used to evaluate the models (Bishop, 2006). However, it is non-trivial to evaluate DNNs based on this single metric. For example, if two networks with the same structure have the similar test accuracy, it is ambiguous which is better.

Robustness against adversarial attacks, measure of the vulnerability, can be an alternative to evaluate DNNs (Szegedy et al., 2013; Goodfellow et al., 2014; Gu & Rigazio, 2014; Huang et al., 2015; Jakubovitz & Giryes, 2018; Yuan et al., 2019; Zhong et al., 2021). Adversarial attacks aim to induce model misprediction by perturbing the input with small magnitude. Most previous works were focused on the way to find adversarial samples by utilizing the model properties such as gradients with respect to the loss function. Given that the adversarial attack seeks to find the perturbation path on the model prediction surface over the input space, robustness can be expressed in terms of the geometry of the model. However, few studies have been performed to interpret the robustness with the concept of the geometric properties of DNNs.

From a geometric viewpoint, the internal properties of DNNs are represented by the boundaries and the regions (Baughman & Liu, 2014). It is shown that the DNNs with piece-wise linear activation layers are composed of many linear regions, and the maximal number of these regions is mathematically related to the expressivity of DNNs (Montúfar et al., 2014; Xiong et al., 2020). As these approaches only provide the upper bound for the expressivity with the same structured model, it does not explain how much information the model actually expresses.

In this work, we investigate the relationship between the internal properties of DNNs and the robustness. In particular, our approach analyzes the internal characteristics from the perspective of the decision boundary (DB) and the decision region (DR), which are basic components of DNNs (Fawzi et al., 2017). To avoid insensitivity of the maximal number of linear regions in the same structure assumption, we propose the novel concept of the *Populated Region Set (PRS)*, which is a set of DRs

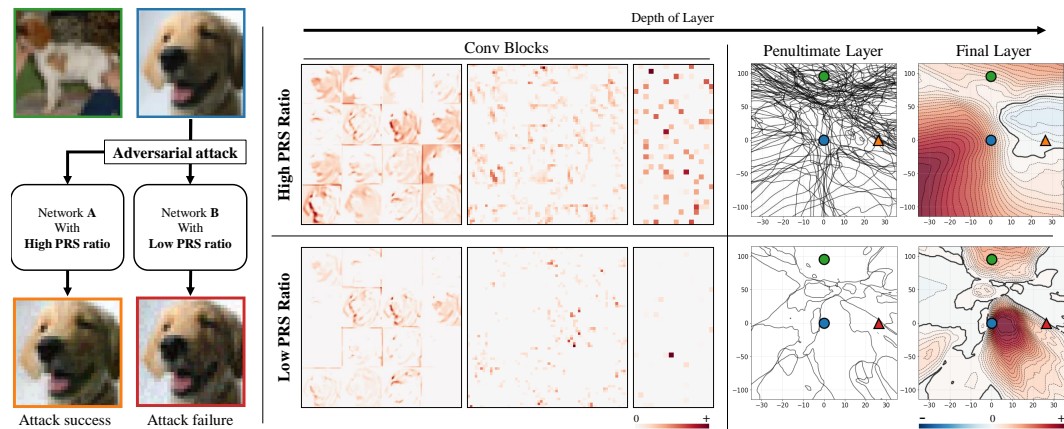

Figure 1: An illustrative comparison of two networks that have different PRS ratios (High and Low) with similar test accuracy. (Left column) Each colored box image corresponds to the colored dot in the right column. The orange/red box image is the perturbed image in each network under an adversarial attack respectively. (Middle column) The network with a low PRS ratio learns more sparse feature representation than that with a high PRS ratio. (Right column) Decision boundaries and regions in the section of input space for the target layer. Contour indicates the logit value for *dog* class.

containing at least one sample included in the training dataset. Since the PRS can be considered as the feasible complexity of the model, we hypothesize that the size of PRS is related to the robustness of network. To validate our hypothesis, we perform systematic experiments with various structures of DNNs and datasets. Our observations are summarized as follows:

- The models with the same structure can have different size of PRS, although they have similar generalization performance. In experiments, we observe that this difference leads to different robustness of the network.

- We empirically show that the size of the PRS is related to the robustness against the adversarial attack. The model with a small size of the PRS tends to show higher robustness compared to that with a large size (in Section 4.1). We further observe that when the model achieves a low PRS ratio, the linear classifier that maps the penultimate features to the logits has high cosine similarity between parameters corresponding to each class (in Section 4.2).

- We verify that the size of intersection of the PRS from the training/test dataset is related to the robustness of model. The model with a high PRS inclusion ratio of test samples has higher robustness than that with a low PRS inclusion ratio (in Section 5).

- We identify that the model with a small size of the PRS learns the sparse feature representation. In quantification, we observe the inversely correlated relationship between the size of the PRS and sparsity of feature representation (in Section 6).

## 2 RELATED WORK

**Adversarial robustness** For the real-world application of DNNs, the adversarial attack, which reveals the vulnerability of DNNs (Goodfellow et al., 2014), is mainly used to validate the reliability of the trained network. As an early stage for adversarial attacks, the fast gradient sign method (FGSM) (Goodfellow et al., 2014) based on the gradient with respect to the loss function and the multi-step iterative method (Kurakin et al., 2016) are proposed to create adversarial examples to change the model prediction with a small perturbation. Recently, many studies on effective attacks in various settings (e.g., white-box or black-box) have been performed to understand the undesirable decision of the networks (Shaham et al., 2018; Madry et al., 2018; Chen et al., 2020). In terms of factors affecting robustness, Yao et al. (2018) provide evidence to argue that training with a large batch size can degrade the robustness of the model against the adversarial attack from the perspective of the

Hessian spectrum. In contrast, Kamath et al. (2019) propose that the model with a constant ratio between the learning rate and batch size does not degrade the model robustness even with a large batch size as it converges to the flatter minima.

**Geometric Analysis inside Deep Neural Networks**   With increasing interest in the expressive power of DNNs, there have been several attempts to analyze DNNs from a geometric perspective (Dauphin et al., 2014; Choromanska et al., 2015). In these studies, the characteristics of the decision boundary or regions formulated by the DNNs are mainly discussed. Montúfar et al. (2014) show that the cascade of the linear layer and the nonlinear activation organizes the numerous piece-wise linear regions. They show that the complexity of the decision boundary is related to the maximal number of these linear regions, which is determined by the depth and the width of the model. Xiong et al. (2020) extend the notion of the linear region to the convolutional layers and show the better geometric efficiency of the convolutional layers. Fawzi et al. (2018) reveal that classification regions in DNNs are topologically connected and the decision boundary of natural images is flat in most directions. It has also been shown that the manifolds learned by DNNs and the distributions over them are highly related to the representation capability of a network (Lei et al., 2018). While these studies highlight the benefits of increasing expressivity of DNNs as the number of regions increases, interpreting the vulnerability of DNNs with the geometry is another important topic. Yang et al. (2020) show that a model with thick decision boundaries induces robustness. Moosavi-Dezfooli et al. (2019) show that a decision boundary with a small curvature acquires the high robustness of the model. These approaches focus on the decision boundaries, while this paper suggests to focus on the decision regions, which are composed by the surrounding decision boundaries.

## 3   INTERNAL PROPERTY OF DNNS

This section describes the internal properties of DNNs from the perspective of decision boundaries (DBs) and regions (DRs). The DBs of the DNN classifier is mainly defined as the borderline between DRs for classification, where the prediction probability of class $i$ and the neighboring class $j$ are the same (Fawzi et al., 2018). To expand the notion of DBs and DRs to the internal feature-level, we re-define the DBs in the classifier that generalizes the existing definition of DBs. We then propose the novel concept of the *Populated Region (PR)* that describes the specific DRs used from the network for training samples. PR is used to analyze the relationship between the trained parameters and the characteristics of networks.

### 3.1   DECISION BOUNDARY AND REGION

Let the classifier with $L$ number of layers be $F(x) = f_L(\sigma(f_{L-1}\sigma(\cdots\sigma(f_1(x))))) = f_{L:1}(x)$, where $x$ is the sample in the input space $\mathcal{X} \subset \mathbb{R}^{D_x}$ and $\sigma(\cdot)$ denotes the non-linear activation function[1]. For the $l$-th layer, $f_l(\cdot)$ denotes the linear operation and $f_{l:1}^i(\cdot)$ denotes the value of the $i$-th element of the feature vector $f_{l:1}(x) \in \mathbb{R}^{D_l}$. We define the DB for the $i$-th neuron of the $l$-th layer.

**Definition 1 (Decision Boundary (DB))** *The $i$-th decision boundary at the $l$-th layer is defined as*

$$B_l^i = \{x | f_{l:1}^i(x) = 0, \quad \forall x \in \mathcal{X}\}.$$

We note that the internal DB $B_l^i$ ($l < L$) divides the input space $\mathcal{X}$ based on the hidden representation of the $l$-th layer (i.e., existence of feature and the amount of feature activation). There are a total of $D_l$ boundaries and the configuration of the DBs are arranged by the training. As input samples in the same classification region are considered to belong to the same class, the input samples placed on the same side of the internal DB $B_l^i$ share the similar feature representation. In this sense, we define the internal DR, which is surrounded by internal DBs.

**Definition 2 (Decision Region (DR))** *Let $\mathbf{V}_l \in \{-1, +1\}^{D_l}$ be the indicator vector to choose positive or negative side of decision boundaries of the $l$-th layer. Then the decision region $DR_{\mathbf{V}_l}$, which shares the sign of feature representation, is defined as*

$$DR_{\mathbf{V}_l} = \{x | sign(f_{l:1}(x)) = \mathbf{V}_l, \quad \forall x \in \mathcal{X}\}.$$

---

[1]Although there are various activation functions, we only consider ReLU activation for this paper.

Figure 1 presents the internal properties for two networks trained on CIFAR-10 with similar test accuracy. The right column in Figure 1 depicts the internal DBs and DRs in the network with a high/low PRS ratio (top and bottom). We randomly select two test images (blue and green box) and generate adversarial images for blue box (orange and purple box) in each network, respectively. We make a hyperplane with these images to visualize the DBs and DRs in the 2D space. We identify that the configuration of DBs and DRs appears to be different, although the two networks have the same structure and similar test accuracy.

## 3.2 POPULATED REGION SET

It is well-studied that the number of DRs is related to the representation power of DNNs (Montúfar et al., 2014; Xiong et al., 2020). In particular, the expressivity of DNNs with partial linear activation function is quantified by the maximal number of the linear regions and this number is related to the width and depth of the structure. We believe that although the maximal number can be one measure of expressity, the trained DNNs with finite training data[2] cannot handle the entire regions to solve the task. To only consider DRs that the network uses in the training process, we devise the train-related regions where training samples are populated more frequently. We define the Populated Region Set (PRS), which is a set of DRs containing at least one sample included in the training dataset. PRS will be used to analyze the relationship between the geometrical property and the robustness of DNNs in a practical aspect.

**Definition 3 (Populated Region Set (PRS))** *From the set of every DRs of the model $f$ and given the dataset $\mathbf{X}$, the Populated Region Set is defined as*

$$PRS(\mathbf{X}, f, l) = \{DR_{\mathbf{V}_l} | x \in DR_{\mathbf{V}_l} \; \exists \; x \in \mathbf{X}, \; \forall \; \mathbf{V}_l \in \{-1, 1\}^{D_l}\}.$$

*We can then define a Populated Region as a union of decision regions in PRS as*

$$PR(\mathbf{X}, f, l) = \cup_{DR \in PRS(\mathbf{X}, f, l)} DR.$$

We note that the size of the PRS is bounded to the size of given dataset $\mathbf{X}$. When $|PRS(\mathbf{X}, f, l)| = |\mathbf{X}|$, each sample in training dataset is assigned to each distinct DR in the $l$-th layer. To compare the PRS of networks, we define the PRS ratio, $\frac{|PRS(f, \mathcal{X}, l)|}{|\mathcal{X}|}$, which measures the ratio between the size of the PRS and the given dataset.

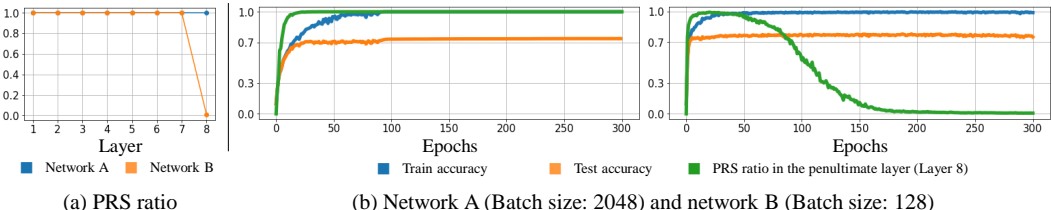

| (a) PRS ratio | (b) Network A (Batch size: 2048) and network B (Batch size: 128) |

Figure 2: (a) The number of PRS for the depth of each layers. (b) Training/Test accuracy and the PRS ratio on the penultimate layer on CNN-6 with batch size 2048/128. We select the networks at the 300th epoch and call these two CNN-6 as Network A and B, respectively, throughout the paper (PRS ratio of Network A: 0.99, and Network B: 0.007).

Figure 2 presents a comparison between two equivalent neural networks (A and B) with six convolution blocks (CNN-6) trained on CIFAR-10 varying only the batch size (2048/128). Figure 2 (a) presents the PRS ratio for the depth of layers in each model at the 300th epoch. We observe that only the penultimate layer ($l = 8$) shows a different PRS ratio. Figure 2 (b) shows that the two networks have largely different PRS ratios with similar training/test accuracy. From the above observation and the fact that the penultimate layers are widely used as feature extraction, we only consider the PRS ratio on the penultimate layer in the remainder of the paper.

**Experimental setups** For the systematic experiments, we select three different structures of DNNs to analyze: (1) a convolutional neural network with six convolution blocks (CNN-6); (2) VGG-16

---

[2]In general, the number of training data is smaller than the maximal number of the linear region.

(Simonyan & Zisserman, 2014); and (3) ResNet-18 (He et al., 2016). We train[3] basic models with fixed five random seeds and four batch sizes (64, 128, 512 and 2048) over three datasets: MNIST (LeCun & Cortes, 2010), F-MNIST (Xiao et al., 2017), and CIFAR-10 (Krizhevsky et al., 2009). For the extensive analysis on the correlation between the PRS ratio and properties of network, we extract candidates from each basic model with the grid of epochs. Then we apply the test accuracy threshold to guarantee the sufficient performance. Finally, we obtain 947 models for analysis. The details for the network architecture and the selection procedure are described in Appendix A-C.

## 4  ROBUSTNESS UNDER ADVERSARIAL ATTACKS

In this section, we perform experiments to analyze the relationship between the PRS ratio and the robustness. We evaluate the robustness of the network using the fast gradient sign method (FGSM) (Goodfellow et al., 2014), basic iterative method (BIM) (Kurakin et al., 2016) and projected gradient descent (PGD) (Madry et al., 2018) method widely used for the adversarial attacks. The untargeted adversarial attacks using training/test dataset are performed for the various perturbation ($\epsilon$).

### 4.1  PRS AND ROBUSTNESS FOR MODELS

First, we compare the two models (Network A and B in Figure 2) with similar test accuracy but different PRS ratio.[4] Figure 3 presents the results of robust accuracy (RA) under the FGSM, BIM (5-step), PGD-20 (20-step), and PGD-100 (100-step) on $L_\infty$. For each step, $\alpha = 2/255$ is adopted. We identify that Network B (low PRS ratio) is more robust than Network A (high PRS ratio) under all adversarial attacks.

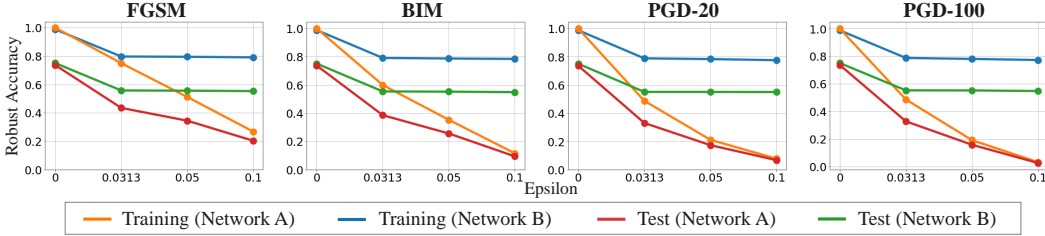

Figure 3: Robust accuracy under various adversarial attacks on networks A and B. The x-axis indicates perturbation $\epsilon$ and the y-axis indicates the training/test robust accuracy under the attack.

As the PGD attack with a 20-step size shows the similar RA compare to a 100-step size, we focus on an analysis under the PGD attack with a 20-step size in the rest of the paper. We measure the PRS ratio and the RA in all models and datasets to verify the relationship between the PRS ratio and the robustness. For the experiments, we take the magnitude of $\epsilon$ as follow: MNIST = 0.3, F-MNIST = 0.1, and CIFAR10 = 0.0313 on $L_\infty$ norm. Figure 4 presents the experimental results according to the model structure under the PGD attack. To quantify the relation, we calculate the slope of the regression line and perform significance test to validate the trend. We also provide the result of RA against the FGSM attack and AutoAttack (Croce & Hein, 2020) in Appendix G. From Figure 4, we identify that the PRS ratio has an inversely correlated relationship with the RA in most cases. The complicated models (VGG-16 and ResNet-18) have lower PRS ratios compared to CNN-6, and show even lower PRS ratios on the simple datasets (i.e., MNIST and F-MNIST).

**PRS and Robustness for Training** We also analyze how the PRS ratio and the RA change during the training. We measure the PRS ratio and the RA under the PGD attack for three different networks trained on CIFAR-10. Figure 5 presents the test accuracy, PRS ratio and RA for each training epoch. Although the test accuracy of each model converges after the 50th epoch, the PRS ratio continues to decrease after the 50th epoch and the RA continues to increase.

---

[3]Cross-entropy loss and Adam optimizer with learning rate $10^{-3}$ is used.
[4]We note that different PRS ratios are obtained by different batch size of Network A (2048) and B (128).

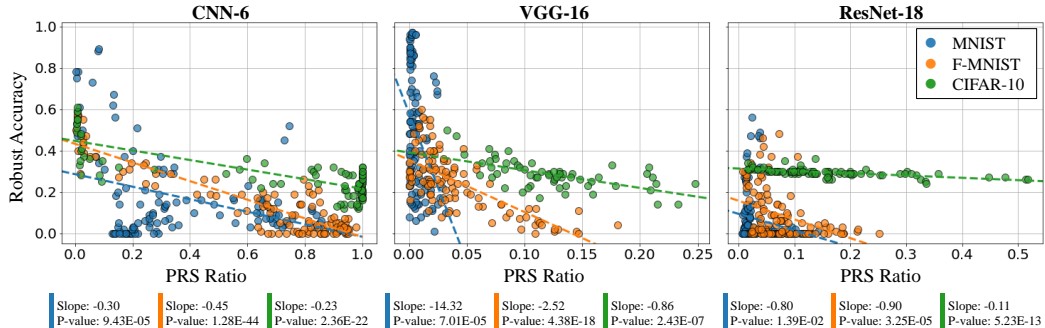

Figure 4: Relationship between the PRS ratio and RA attacked by PGD method in various models and datasets. The colored dots are for the independent models described in Appendix A. The colored dashed lines indicate the trend for each dataset.

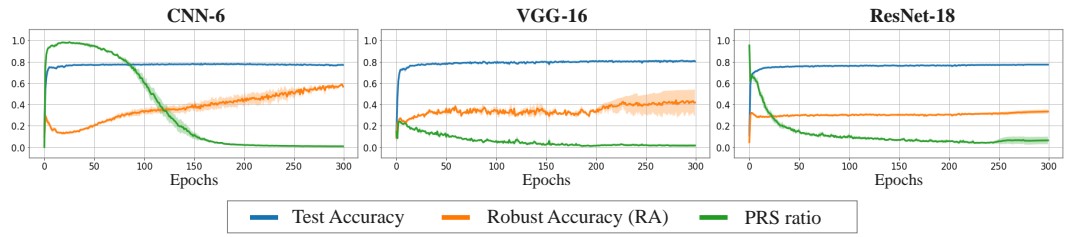

Figure 5: PRS ratio and RA for training epochs on three different networks with CIFAR-10. The blue line indicates the test accuracy, the orange line indicates the RA, and the green line indicates the PRS ratio. The shaded lines depict the standard deviation for five fixed random seeds.

## 4.2 PRS AND FINAL LAYER

From the observations above, we empirically confirm that the PRS ratio is related to the robustness against adversarial attacks. In order to investigate the evidence that the low PRS ratio causes robustness for the gradient-based attack, we perform an additional analysis of failed attack samples. In the gradient-based attack, as the magnitude of the gradient is a crucial component to success, we first count the ratio of the zero gradient samples in the failed attack samples. The Figure 6 (a) shows the ratio of success samples (light green bar), failure samples with non-zero gradient (blue bar) and zero gradient (red bar) in all samples. We note that the failed attack samples with non-zero gradients maintain the index of the largest logit as the true class after attack. To analyze the reason of failure, we examine the change of the logits under the adversarial attack. This change is shown in Figure 6 (b). To clarify the difference of the change of the logits between Network A and B, we select the examples of successful attack on Network A but failed attack on Network B. In Network B, the logits move on almost parallel direction, which causes the predicted label to be maintained as the true class.

To explain the parallel change of the logit vector, we hypothesize that the DBs corresponding to each class node have similar configuration in the input space. However, it is intractable to measure the similarity between DBs in the entire network due to the highly non-linear structure and the high dimensional input space. To simplify our hypothesis, we only measure the cosine similarity between the parameters which map the features on the penultimate layer to logits (i.e., final layer). Figure 7 presents that the similarity matrices for Networks A and B. When we compare the matrix between the two models at the 300th epoch, we identify that Network B (low PRS ratio) has greater cosine similarity between each parameter in the final layer. We note that the cosine similarity between each parameter in the final layer can be considered as the degree of parallelism for the normal vectors in the linear classifier. We also confirm that the decrease of the PRS ratio is aligned with the increase of the similarity of parameters in the Figure 7 (b), when we consider the graph in Figure 2. To verify the relationship between PRS ratio and the cosine similarity between the parameters in the final layer, we measure the PRS ratio and the cosine similarity between each parameter in all models. Figure 8 shows the experiment results grouped according to the model structure. We identify that

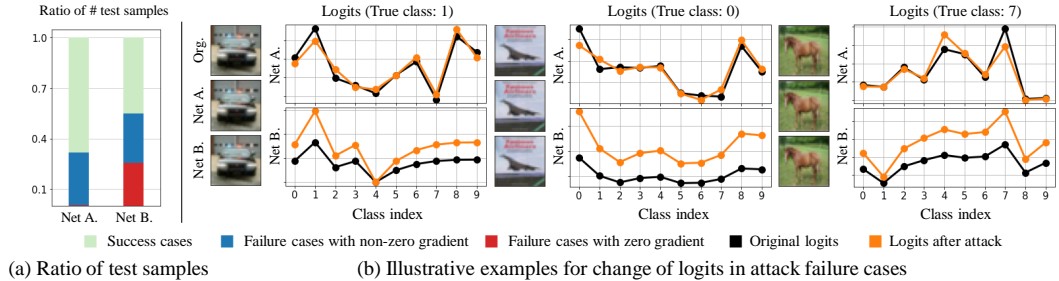

(a) Ratio of test samples

(b) Illustrative examples for change of logits in attack failure cases

Figure 6: (a) Comparison of the ratio of the zero gradient in the failure attack for the test samples under the PGD attack on $L_\infty$ with $\epsilon = 0.0313$ and 20-step size (Network A and B). (b) The illustrative examples of attacked samples on Network A and B which is failed on B, and the corresponding logits before/after the attack. After the attack, the logits move on almost parallel direction with the original logits in Network B. More examples are provided in Appendix C.

the PRS ratio has an inverse correlation for the cosine similarity between each parameter in the final layer.

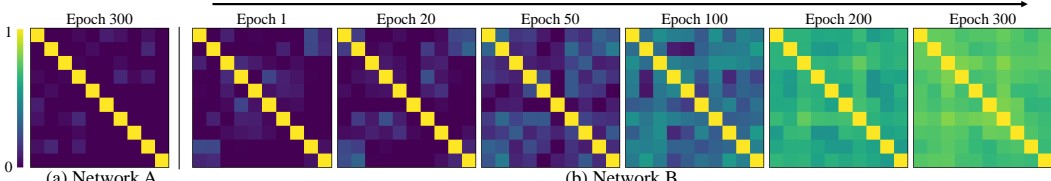

(a) Network A

(b) Network B

Figure 7: (a) Cosine similarity matrix for the final layer on Network A. (b) Similarity matrix for epochs on Network B. As the epoch increases, the cosine similarity for each parameter increases.

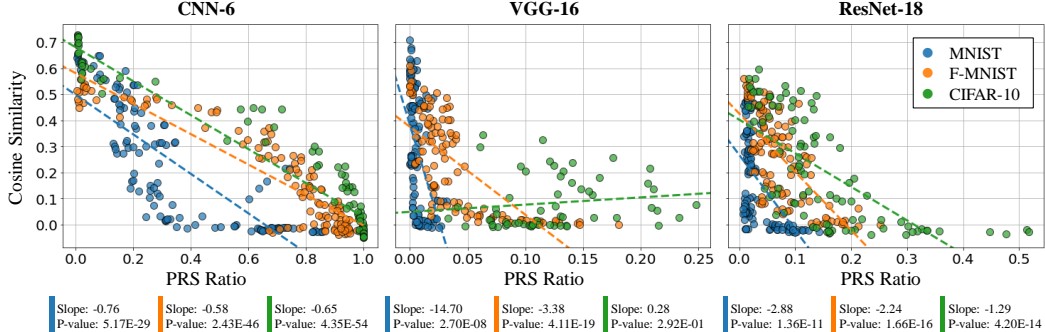

Figure 8: Relationship between the PRS ratio and the cosine similarity in various models and datasets. The colored dots represent the independent models. The colored dashed lines indicate the trends for each of the datasets.

## 5 PRS AND TEST DATASET

Regarding the model as a mapping function from the input space to the feature space, handling unseen data in a known feature domain is significant with regard to the generalization performance of the model. Hence, if the majority of samples from the test dataset are assigned to the training PR, the model can be considered to learn the informative and general concept of feature mapping. For example, if the arbitrary test sample is mapped to the training PR, we expect that a similar decision will appear. However, it is non-trivial to guess which type of decision will appear when the test sample is mapped to out of the training PR.

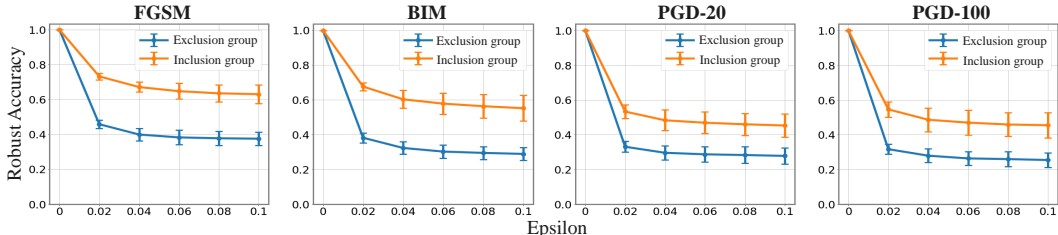

Figure 9: Test accuracy under adversarial attacks for inclusion/exclusion groups for CNN-6 on CIFAR-10 for five fixed random seeds. The blue/orange line indicates the exclusion/inclusion groups, respectively. The exclusion group is shown to be more vulnerable under adversarial attacks.

To investigate the differences between the test samples which are included/excluded in the training PR, we evaluate the test accuracy under adversarial attack for each group. For a controlled comparison, we divide both the inclusion and exclusion groups with 1k correctly predicted test samples.

Figure 9 shows the RA under the FGSM, BIM with a 5-step size, and the PGD with 20 and 100 step size on $L_\infty$. Although the accuracy of each test group decreases as the epsilon becomes larger, we observe that the inclusion group is more robust against all types of attacks compared to the exclusion group. We also provide the result of other networks and datasets in Appendix I.

Figure 10 presents the PRS ratio and the inclusion ratio of the test samples for the training PR. We compute the inclusion ratio as the ratio of the test samples mapped to the training PR. In Figure 10, we identify that the PRS ratio and the inclusion ratio exist in inversely correlated relationship. As we previously verify that the included test samples show high robustness, we empirically confirm that the low PRS ratio is related to the robustness under adversarial attacks.

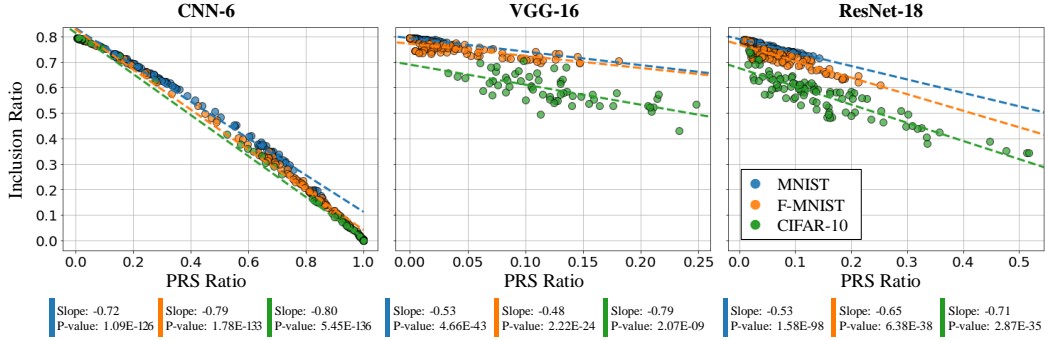

Figure 10: Relationship between the PRS ratio and the inclusion ratio for CNN-6, VGG-16 and ResNet-18 on the various datasets. The colored dashed lines indicate the trend for each dataset.

# 6 PRS AND TRAINED FEATURES

This section explores the trained features of the models with the different PRS ratios. First, we visualize the feature maps directly for each depth of layers. Figure 11 shows illustrative examples of feature maps. We identify that the model with the low PRS ratio learns more sparse features compared to the model with the high PRS ratio. As the sparse features are considered as an independent and informative representation (Ranzato et al., 2007; Lee et al., 2007), if the PRS ratio can cause sparse feature representation in various cases, we conclude that the PRS ratio is related to the informative features. To verify our hypothesis, we measure the trend between the PRS ratio and the average sparsity for each network. The average sparsity of the model is calculated by taking the average of the ratio of zero-valued features over the training dataset for all layers.

Figure 12 shows the relationship between the PRS ratio and the average sparsity in all cases. We identify that CNN-6 and VGG-16 show an inversely correlated relationship throughout the dataset, but it is difficult to find a clear relationship for ResNet-18. We conjecture that (1) skip connection, or

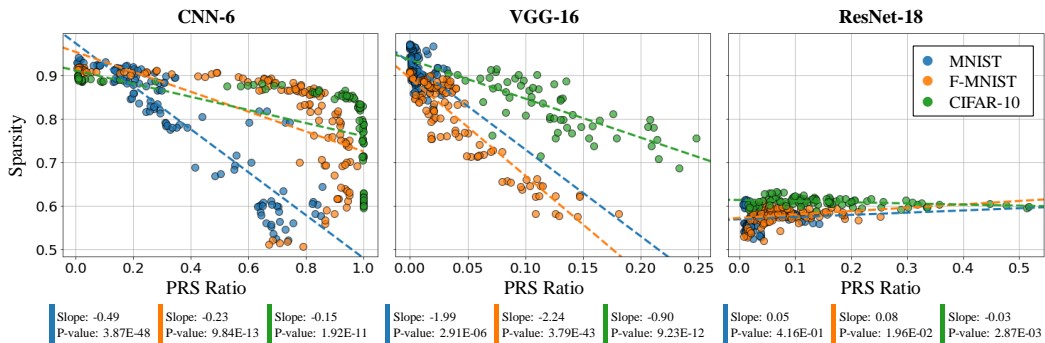

Figure 11: Visualization of feature maps for CNN-6 trained on CIFAR-10. (First row) Feature maps for Network A. (Second row) Feature maps for Network B.

| Slope: -0.49 | Slope: -0.23 | Slope: -0.15 | Slope: -1.99 | Slope: -2.24 | Slope: -0.90 | Slope: 0.05 | Slope: 0.08 | Slope: -0.03 |
| P-value: 3.87E-48 | P-value: 9.84E-13 | P-value: 1.92E-11 | P-value: 2.91E-06 | P-value: 3.79E-43 | P-value: 9.23E-12 | P-value: 4.16E-01 | P-value: 1.96E-02 | P-value: 2.87E-03 |

Figure 12: Relationship between the PRS ratio and the average sparsity for CNN-6, VGG-16 and ResNet-18 on the various datasets. The colored dashed lines indicate the trend for each dataset.

(2) batch normalization can cause this phenomenon. To investigate the global feature representations further, we visualize the projected feature space using t-SNE (van der Maaten & Hinton, 2008) for the training dataset. In Figure 6, we can identify that the samples in low PRS ratio cases are more densely located in each class, while the samples show widely spread pattern in each class in high PRS ratio cases.

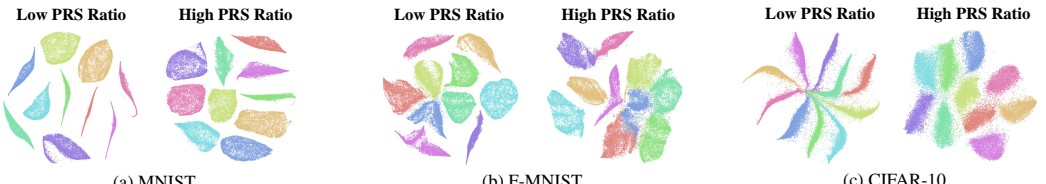

(a) MNIST       (b) F-MNIST       (c) CIFAR-10

Figure 13: Visualization of the projected feature space with t-SNE for CNN-6 trained with various datasets. The colored code indicates the class. The samples in low PRS ratio cases are more densely located in each class compared to the samples in high PRS ratio.

## 7 CONCLUSION

In this work, we perform an empirical study to analyze the internal properties of DNNs, which affect the robustness under adversarial attacks. We propose the novel concept *populated region set* to derive the relationship between the internal properties of DNNs and robustness in a practical setting. From systematic experiments for the proposed concept, we suggest that the PRS ratio is related to the robustness of DNNs and provide empirical evidence of this relationship: (1) The network with the a low PRS ratio shows high robustness against the gradient-based attack compared to the network with a high PRS ratio. In particular, the model with a low PRS ratio has a higher degree of parallelism for the parameters in the final layer, which can support robustness. (2) The network with a low PRS ratio includes more test samples in the training PR. We empirically verify that this inclusion ratio is related to robustness from the observation that included test samples are more robust than excluded test samples. (3) The network with a low PRS ratio learns sparse feature representation to solve tasks. Moreover its global feature representation is more intensive than the network with a high PRS ratio.

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
