# OpenReview forum: "Empirical Study of the Decision Region and Robustness in Deep Neural Networks"
_ICLR.cc/2022/Conference — ICLR 2022 Submitted_

### Official Review · Reviewer_5u8a · 2021-10-29

**Correctness:** 3
**Technical Novelty And Significance:** 2
**Empirical Novelty And Significance:** 2
**Recommendation:** 6
**Confidence:** 4

**Main Review:**

Strength:
The idea of populated regions is interesting, and should be further investigated. Especially, how test samples that fall into them are more robust could be a good direction to study in relation to designing more robust models.
The authors investigate PRS from many different directions and show how it is correlated to other important properties across 3 datasets and 3 model types.

Weaknesses:
The paper has big problems with clarity. Two networks, one with a large PRS (network A) and one with a small PRS (network B) are used to explore how the size of the PRS is related to other properties, but neither in the main paper nor in the appendix is it explained how they differ from each other. Is it just a different random seed or are they trained differently? In figure 2 it looks like network B reaches a high train accuracy significantly faster than network A.

In figure 4, 8 and 10 we see scatter plots of lots of models trained but again it is not clear how they are different. The appendix lists clean accuracy for models in table 2,3 and 4 but it only shows 4 different batch sizes and 5 different random seeds. The scatter plots clearly have more than these 20 combinations per dataset.
Additionally, I don't understand how the confidence intervals for the accuracies in the appendix tables are calculated, since the accuracies are for a fixed random seed so each run should lead to the same result.

The paper never says what norm is used for the adversarial attacks. From the epsilon values used I assume it is l_2. Do the results hold for l_inf?

I'm also not sure how useful the PRS metric is. The authors give the example that it is hard to choose between models with similar clean accuracies. But in that case I don't see the advantage of using the PRS ratio instead of directly using robust accuracy. Since there is no explanation on why some models have different PRS ratio it also doesn't give any hints at how one could deliberately train models to achieve a lower PRS ratio and therefore be more robust.

After comparing FGSM and iterative attacks (both targeted and untargeted) in figure 3 they use FGSM (untargeted) for the rest of the paper because the difference between the networks is the biggest. I don't think this is a good choice since the comparison clearly shows it is the weakest attack. It would a lot more interesting to see if the results hold for even stronger attacks like PGD or auto-attack[1].

There are also a lot of grammatical errors in the paper which make it hard to read and understand. There are too many to list here so I would encourage the authors to have someone proofread the draft again.

Other comments & questions:

Since PRS size is correlated with robustness do models trained adversarially have very small PRS compared to when we use standard training?

On page 1: "We further observe that the model with the small size of PRS has the relatively parallel parameters compared to the large one in the final layer."
I could only understand what is meant here after reading the section at the end of the paper.

Figure 2: You should make clear that the left and middle plot are only showing the PRS ratio of layer 8.

Figure 6: Shouldn't the ratio of test samples with zero and non-zero gradient sum up to 1 for both networks? I don't understand what the bars show exactly. Additionally, instead of two images for Network B ist would be more interesting two show how a successful attack for Network A changes the
logits.

Figure 13: I'm not sure what "more intensive feature representation" means here. I find it hard to make any qualitative claims from the plots. For CIFAR-10 you could even argue that the high PRS ratio shows better separation of the classes while they all melt together in the middle for the low PRS ratio model.

[1]"Reliable evaluation of adversarial robustness with an ensemble of diverse parameter-free attacks"
Francesco Croce, Matthias Hein
ICML 2020

**Summary Of The Paper:**

The paper proposes a new metric, the size of the populated region set (PRS), as an explanation for models with similar clean accuracies reaching very different accuracies under adversarial attacks. PRS is the set of decision regions that have training examples in them. After introducing and defining populated regions show that PRS in the penultimate layer is inversely correlated with robust accuracy. This is shown to hold across MNIST, F-MNIST and CIFAR10 and for 3 different model architectures (simple CNN, VGG and ResNet). They use two example networks with high and low PRS to show how PRS evolves during training and that the higher resistance to adversarial attacks might stem from the neurons in the last layer having similar decision boundaries in input space.
Test samples that also fall into the PRS are shown to be more robust and models with small PRS have more test samples in them than models with large PRS. Additionally, it is shown that models with small PRS produce more sparse features.

**Summary Of The Review:**

While the concept is interesting the paper doesn't provide enough details about how the models were trained or about the adversarial attacks. I'm not convinced the newly introduced metric will be useful without a deeper evaluation, especially explaining why similar models differ drastically in PRS size. The paper is hard to read and understand as there are a lot of grammatical errors throughout.

Given all these factors I don't think this paper should be accepted.

---

> ### Author Response · Authors · 2021-11-16
> **Response to Reviewer 5u8a (5/5)**
>
> ---
>
>
> **Q10. Figure 13: I'm not sure what "more intensive feature representation" means here. I find it hard to make any qualitative claims from the plots. For CIFAR-10 you could even argue that the high PRS ratio shows better separation of the classes while they all melt together in the middle for the low PRS ratio model.**
>
> ---
>
> - We agree that the inappropriate term "intensive" causes the ambiguity of the contents. We modified the description to clarify the contents. Our intention was that a network with a low PRS ratio tends to show more densely located samples per each class in the feature space on the penultimate layer. We provided the t-SNE figures as an method to visualize this property.
>
> - To support our hypothesis, we also provide the quantitative experiment to measure denseness of the feature representations in each class over the networks with different PRS ratio. In experiment, we measure the ratio of same class samples in the feature space based on $K$ nearest neighbors. Let the training dataset $\mathbf{X}$ and data pair $(x_i,y_i)\in \mathbf{X}$. The denseness for the given data pair $(x,y)$ is defined as,
>
> $$D(\mathbf{X},K)=\frac{1}{|\mathbf{X}|}\sum_{x,y \in \mathbf{X}}\frac{1}{K}\sum_{i\in NN_K(x, \mathbf{X})} \mathbf{I}(y_i=y), \quad $$
>
> where $NN_K(x,\mathbf{X})$ returns indices of $K$ nearest neighbors for feature representation of the given input $f_{L-1:1}(x)$ from training dataset $\mathbf{X}$ and $\mathbf{I}(\cdot)$ is the indicator function.
>
> - Table 1 presents the average denseness for various random seeds mentioned in Table 2 of Appendix A with various $K$. The experiment is performed on CNN-6 with low/high PRS ratio (we use trained model with batch size of 128 and 2048 to control the PRS ratio). We identify that the low PRS ratio cases (a1, b1, c1) have higher densness compared to the high PRS ratio cases (a2, b2, c2) in almost $K$. For MNIST dataset, it is difficult to discriminate the difference of denseness along the magnitude of PRS ratio.
>
> **Table 1.** Average of denseness in feature space for models with a low PRS ratio and a high PRS ratio.
> We trained CNN-6 on three datasets, (a) MNIST, (b) F-MNIST and (c) CIFAR-10.
> Odd rows (a1, b1, c1) indicate the model with a low PRS ratio and even rows (a2, b2, c2) indicate the model with a high PRS ratio.
>
> |      |  PRS ratio  |      10     |      50     |     100     |     200     |     300     |
> |:----:|:-----------:|:-----------:|:-----------:|:-----------:|:-----------:|:-----------:|
> |(a1) | 0.035${\pm}$0.029 | **0.999${\pm}$0.001** | **0.998${\pm}$0.001** | 0.997${\pm}$0.001 | 0.996${\pm}$0.002 | 0.995${\pm}$0.002 |
> |(a2) | 0.669${\pm}$0.088 | 0.999${\pm}$0.001 | **0.998${\pm}$0.001** | **0.998${\pm}$0.001** | **0.997${\pm}$0.002** | **0.996${\pm}$0.002** |
> |(b1) | 0.022${\pm}$0.007 | **0.981${\pm}$0.006** | **0.972${\pm}$0.008** | **0.966${\pm}$0.009** | **0.957${\pm}$0.010** | **0.949${\pm}$0.011** |
> |(b2) |0.946${\pm}$0.018 | 0.969${\pm}$0.003 | 0.955${\pm}$0.003 | 0.948${\pm}$0.004 | 0.938${\pm}$0.004 | 0.932${\pm}$0.004 |\hline
> |(c1)  | 0.006${\pm}$0.000 | **0.945${\pm}$0.006** | **0.920${\pm}$0.007** | **0.903${\pm}$0.007** | **0.877${\pm}$0.007** | **0.857${\pm}$0.007** |
> |(c2) | 1.000${\pm}$0.000 | 0.939${\pm}$0.007 | 0.904${\pm}$0.008 | 0.883${\pm}$0.009 | 0.855${\pm}$0.010 | 0.836${\pm}$0.011|
>
> ---
>
> We sincerely thank you for the insightful comments since it helped us to further improve the discussions in the paper.
> Please let us know if there is anything else we should address, or misunderstood.
>
> ---
>
> **Reference**
>
> [1] Ilyas, Andrew, et al. “Adversarial examples are not bugs, they are features.", NeurIPS 2019
> [2] Yao, Zhewei, et al. “Hessian-based analysis of large batch training and robustness to adversaries.”, NeurIPS 2018
> [3] Benz et al. “Batch Normalization Increases Adversarial Vulnerability and Decreases Adversarial Transferability: A Non-Robust Feature Perspective.”, ICCV 2020
> [4] Shaeiri, Amirreza et al. “Towards Deep Learning Models Resistant to Large Perturbations.”, ICLR 2018
> [5] Croce, Francesco et al. “Reliable evaluation of adversarial robustness with an ensemble of diverse parameter-free attacks.”, ICML 2020

---

> > ### Comment · Reviewer_5u8a · 2021-11-19
> > **Answer**
> >
> > Thanks again for the clarification and quantization of the claim.

---

> ### Author Response · Authors · 2021-11-16
> **Response to Reviewer 5u8a (4/5)**
>
> ---
>
>
> **Q6. Since PRS size is correlated with robustness do models trained adversarially have very small PRS compared to when we use standard training?**
>
> ---
>
> - Thank you for the insightful comment. To verify the relationship between adversarial training (AT) and the PRS ratio, we perform the comparison between standard training (ST) and AT. We first train the ResNet-18 with CIFAR-10 dataset using standard training as the pre-trained model (0th - 150th epoch). We then train two networks from this pre-trained model using (1) ST, and (2) AT based on the PGD on $L_\infty$ with $\epsilon=0.0313$. After training, we compare the PRS ratio and the robust accuracy against PGD attack. Table 1 presents the PRS ratio and the robust accuracy in the training. We observe that the PRS ratio drops and the robust accuracy increases at the same epoch (160th epoch), while those of ST almost maintain. We also provide this result in Figure 18 of Appendix H.
>
> **Table 1.** Comparison of PRS and robust accuracy against PGD attack between the ST AT on the same pretrained model.
>
> |      Epoch     |   100 |   110 |   120 |   130 |   140 |   150 |   160 |   170 |   180 |   190 |   200 |
> |:--------------:|------:|------:|------:|------:|------:|------:|------:|------:|------:|------:|------:|
> |  ST PRS Ratio  | 0.518 | 0.494 | 0.470 | 0.457 | 0.442 | 0.412 | 0.353 | 0.349 | 0.346 | 0.345 | 0.343 |
> |  AT PRS Ratio  | 0.518 | 0.494 | 0.470 | 0.457 | 0.442 | 0.412 | **0.211** | **0.181** | **0.211** | **0.208** | **0.217** |
> | ST Robust Acc. | 0.211 | 0.215 | 0.231 | 0.233 | 0.228 | 0.239 | 0.247 | 0.247 | 0.246 | 0.247 | 0.246 |
> | AT Robust Acc. | 0.211 | 0.215 | 0.231 | 0.233 | 0.228 | 0.239 | **0.401** | **0.408** | **0.420** | **0.423** | **0.425** |
>
>
>
> **Q7. On page 1: "We further observe that the model with the small size of PRS has the relatively parallel parameters compared to the large one in the final layer." I could only understand what is meant here after reading the section at the end of the paper.**
>
>
> ---
>
> - To clarify the statement, we have changed the notion to clearly represent the parameters between the features on the penultimate layer and the logit of each class. We also denote the Section number for each observation in the page 2 to increase accesibility for readers.
>
> ---
>
>
> **Q8. Figure 2: You should make clear that the left and middle plot are only showing the PRS ratio of layer 8.**
>
> ---
>
> - Thanks for pointing out the ambiguity. We denote what each colored line refers in the revised figure.
>
> ---
>
> **Q9. Figure 6: Shouldn't the ratio of test samples with zero and non-zero gradient sum up to 1 for both networks? I don't understand what the bars show exactly. Additionally, instead of two images for Network B ist would be more interesting two show how a successful attack for Network A changes the logits.**
>
> ---
>
> - From the all test samples, we divided the cases of attack into (1) success cases, (2) failure cases with the zero gradient and (3) failure cases with the non-zero gradient. In Figure 6, we only included the failure cases (2 and 3), which may not summed up to 1. We agree the ambiguity which you mentioned, and revise the bar plot to include the success case, so that the total ratios can be 1.
>
> - In response to your additional comment, we attached the examples of successful attack on Network A but failed attack on Network B. From the examples, we identify more clear difference between network A and B. we also attached more examples in Appendix C. Thanks again for constructive advice.

---

> > ### Comment · Reviewer_5u8a · 2021-11-19
> > **Answer**
> >
> > Thanks for the clarifications and additional experiments, these are very helpful.

---

> ### Author Response · Authors · 2021-11-16
> **Response to Reviewer 5u8a (3/5)**
>
> ---
>
>
> **Q5. After comparing FGSM and iterative attacks (both targeted and untargeted) in figure 3 they use FGSM (untargeted) for the rest of the paper because the difference between the networks is the biggest. I don't think this is a good choice since the comparison clearly shows it is the weakest attack. It would a lot more interesting to see if the results hold for even stronger attacks like PGD or auto-attack.**
>
> ---
>
> - We measure the robust accuracy for stronger adversarial attack methods over the networks with different PRS ratio. We consider the Projected Gradient Decent (PGD) attack [4], Auto Attack (AA) [5] which you mentioned. For the experiments, we adopt 20-step and take the magnitude of $\epsilon$ as follow: MNIST=0.3, F-MNIST=0.1, and CIFAR10=0.0313 on $L_\infty$-based attack. The used models are selected at the 200th epoch with random seed 907.
>
> - Table 1 indicates the robust accuracy for various architecture and dataset with different batch size. We identify that the observed relationship between PRS ratio and robustness is valid for the stronger attack in almost cases.
>
>
> **Table 1.** Robust accuracy for various attack methods (FGSM, PGD, and AA).
>
> |   Model   | Dataset | Batch Size | PRS Ratio |  Clean |  FGSM  |   PGD  |   AA   |
> |:---------:|:-------:|:----------:|:---------:|:------:|:------:|:------:|:------:|
> |           |  MNIST  |     128    |   0.201   | 99.46  | **83.54**  | **39.79**  |  0.00  |
> |           |         |    2048    |   0.679   | 99.39  | 37.11  | 11.81  |  0.00  |
> |   CNN-6  |  fMNIST |     128    |   0.195   | 92.24  | **52.63**  | **34.05**  |  0.00  |
> |           |         |    2048    |   0.977   | 91.05  | 28.39  |  2.67  |  0.00  |
> |           | CIFAR10 |     128    |   0.022   | 77.60  | **44.71**  | **41.17**  | **23.57**  |
> |           |         |    2048    |   1.000   | 75.66  | 40.10  | 25.92  |  7.48  |
> |           |  MNIST  |     128    |   0.001   | 99.18  | 54.22  | 36.36  |  0.00  |
> |           |         |    2048    |   0.024   | 99.35  | **55.59**  | **36.68**  |  0.00  |
> |   VGG-16  |  fMNIST |     128    |   0.012   | 92.66  | **45.09**  | **30.60**  | **14.28**  |
> |           |         |    2048    |   0.060   | 92.44  | 41.31  | 17.14  |  7.99  |
> |           | CIFAR10 |     128    |   0.012   | 77.90  | **44.40**  | **33.68**  | **21.08**  |
> |           |         |    2048    |   0.106   | 75.05  | 39.92  | 30.03  | 14.22  |
> |           |  MNIST  |     128    |   0.013   | 99.34  | **51.15**  | **24.88**  |  0.00  |
> |           |         |    2048    |   0.078   | 99.37  |  0.40  |  0.00  |  0.00  |
> | ResNet-18 |  fMNIST |     128    |   0.027   | 90.75  | **31.56**  | **19.94**  |  **0.60**  |
> |           |         |    2048    |   0.078   | 90.29  | 26.55  | 12.98  |  0.00  |
> |           | CIFAR10 |     128    |   0.073   | 75.40  | **40.27**  | **29.87**  | **28.52**  |
> |           |         |    2048    |   0.334   | 72.08  | 32.61  | 24.51  | 23.45  |
>
> - For deeper analysis on PGD, we perform the experiment mentioned in the main paper for verification of the significance. The correlation scatter plots are attached in Figure 17 of Appendix G. The Table 2 indicates the calculated results for slopes and correspondig P-value of significance test. We identify that the PRS ratio is inversely correlated to the robust accuracy in most cases. However, we observe the weak correlation in CNN-6 with some dataset (MNIST and CIFAR-10). We conjecture that this phenomenon is caused by the shallow architecture of CNN-6, because we observe similar weak correlation for robustness over FGSM attack (Figure 4 in the main paper).
>
> **Table 2.** The slope of regression line and P-value for significance test.
>
> |   Model  | Dataset |  Slope  |  P-value |
> |:--------:|:-------:|:-------:|:--------:|
> | 　       | MNIST   | -0.30   | 9.43E-05 |
> | CNN-6   | fMNIST  | -0.45   | 1.28E-44 |
> | 　       | CIFAR10 | -0.23   | 2.36E-22 |
> | 　       | MNIST   | -14.32  | 7.01E-05 |
> | VGG-16    | fMNIST  | -2.52   | 4.38E-18 |
> | 　       | CIFAR10 | -0.86   | 2.43E-07 |
> | 　       | MNIST   | -0.80   | 1.38E-02 |
> | ResNet-18 | fMNIST  | -0.90   | 3.25E-05 |
> | 　       | CIFAR10 | -0.11   | 5.23E-13 |

---

> > ### Comment · Reviewer_5u8a · 2021-11-19
> > **Answer**
> >
> > Thanks for the new experiments using PGD and AA. I'm satisfied that these show that the improvement is real and not just an artefact of the attack method.

---

> ### Author Response · Authors · 2021-11-16
> **Response to Reviewer 5u8a (2/5)**
>
> ---
>
>
> **Q4-1. I'm also not sure how useful the PRS metric is. The authors give the example that it is hard to choose between models with similar clean accuracies. But in that case I don't see the advantage of using the PRS ratio instead of directly using robust accuracy.**
>
> ---
>
>
> - In our best knowledge, because previous works mainly focus on the identification for the successful attack samples to quantify the robustness of the network, it is still curious which samples are vulnerable against adversarial attacks.
>
> - We think that the proposed concept, PRS, can provide an insight for this question. In our work, we categorize the successful attack samples with two groups: (1) samples included in the PRS, and (2) excluded from the PRS. In particular, we empirically show that the exclusion group is more vulnerable to the gradient-based adversarial attack (Section 5 in the main paper and Appendix I).
>
> - This categorization can be an informative tool to trace reasons for the vulnerability of DNNs. For example, the inclusion group may be related to non-robust features [1]. On the other hand, the exclusion group may come from the weak generalization. We believe that this analysis can support to understand the adversarial vulnerability of DNNs and provide the geometrical interpretation of robustness.
>
>
> ---
>
> **Q4-2. Since there is no explanation on why some models have different PRS ratio it also doesn't give any hints at how one could deliberately train models to achieve a lower PRS ratio and therefore be more robust.**
>
> ---
>
> - We perform an ablation study on the factors which affects the PRS ratio and the results are provided in Appendix F.
>
> - First, we identify the relationship between the PRS ratio and batch size (BS) which is one of the factors we used to make the candidates in section A. Figure 16 (a) of Appendix F shows the PRS ratio and test accuracy for training epochs in different BS (64, 128, 512, 2048) and different networks (CNN-6, VGG-16, ResNet-18) on CIFAR-10. We observe that BS is proportional to PRS ratio (i.e., The large BS causes the high PRS ratio). Previous work [2] provides that training with large batch size can degrade the robustness of the model against the adversarial attack, which is aligned with our observation.
>
> - To further investigate other factors, we select two training techniques used in general; (1) Drop out (DO), and (2) Batch normalization (BN). In order to minimize the other influences such as the structural characteristics (e.g., Skip Connection of ResNet), we adopt CNN-6 as the base model in ablation study. Figure 16 (b) indicates the PRS ratio and test accuracy for training epochs according to the existence of DO ($p=0.2$) over various BS on CIFAR-10. We find that DO tends to delay the decrease of PRS ratio in the cases of BS 64 and 128. However, if the network has high PRS ratio (BS 512 and 2048), DO does not affect to change of the PRS ratio.
>
> - Figure 16 (c) represents the result of the training with the existence of BN for various BS on CIFAR-10. We find that the models with BN have the high PRS ratio in all cases. When we consider the relation between the PRS ratio and robustness, BN can be considered as the factor which causes adversarial vulnerability. Previous work [3] describes the negative effect of BN on the robustness.

---

> > ### Comment · Reviewer_5u8a · 2021-11-19
> > **Answer**
> >
> > Thanks for the additional results in the appendix these are very helpful. I think what this paper is still missing is a theory on why other parameters like batch size, drop out or batch normalisation are correlated with PRS. The authors e.g cite previous research linking batch size to robustness. What advantage does PRS bring to this analysis? Does it help to find other parameters that could improve robustness.
> >
> > My main point is: The metric we actually care about is the robust accuracy and we can calculate it fairly easily. What is the advantage of caculating PRS over the robust accuracy? Is it faster? Less easy to fool (e.g. through gradient obfuscation) or does it gives us a better idea how to make models more robust?

---

> ### Author Response · Authors · 2021-11-16
> **Response to Reviewer 5u8a (1/5)**
>
> We sincerely appreciate your time and efforts in reviewing our paper, as well as the constructive comments. We have revised the manuscript by faithfully reflecting your comments. We respond to your comments below.
>
>
> ---
>
> **Q1. What is difference between network A and B (only random seed is different?)**
>
> ---
>
> - We are sorry for skipped description for network A and B. We train the network with different batch size (A:2048, B:128) and other hyper-parameters are fixed; (1) random seed: 375, (2) epoch: 300, (3) learning rate: 1e-3, and (4) optimizer: Adam optimizer.
>
> ---
>
> **Q2. In figure 4, 8 and 10 we see scatter plots of lots of models trained but again it is not clear how they are different. The appendix lists clean accuracy for models in table 2,3 and 4 but it only shows 4 different batch sizes and 5 different random seeds. The scatter plots clearly have more than these 20 combinations per dataset. Additionally, I don't understand how the confidence intervals for the accuracies in the appendix tables are calculated, since the accuracies are for a fixed random seed so each run should lead to the same result.**
>
> ---
>
> - We agree that our description made you confused, so provide the details for the used models in Table 2 of Appendix A. As you mentioned in the question, we train 20 combinations (we noted as the basic models) with the 5 random seeds and 4 batch size for each dataset and architecture of model. To obtain more various PRS ratio with similar test performance, we extract candidates from these basic models. For systematic extraction, we set the grid of epochs ([25,50,75,100,200,300]) and collect the corresponding trained parameters as the independent candidates. We apply the test accuracy ($acc$) threshold (MNIST: $98\%\leq acc$, F-MNIST: $90\%\leq acc \leq93\%$) CIFAR-10: $72\%\leq acc \leq78\%$) for the candidates to bound the simliarity of the performance. Table 2 of Appendix A indicates the all selected/rejected candidates for experiments.
>
> - We also attach the illustrative example for extracting procedure of candidates in Figure 14 of Appendix A.
>
>
> - The mentioned confidence intervals described in Table 3 of Appendix B are calculated in the selected epochs. For example, the mean and standard deviation for  VGG-16 with MNIST (random seed 375 and batch size 64), we average the test accuracy for the models corresponding 25, 70, 100, 200, and 300 epochs (blue box in Table 2 and Table 3 of Appendix). In short, the scatters in Figure 4, 8, 10, and 12 (in the main paper) correspond to each selected candidate and scatters differ in (1) random seeds, (2) batch size, and (3) epoch.
>
> ---
>
> **Q3. The paper never says what norm is used for the adversarial attacks. From the epsilon values used I assume it is l_2. Do the results hold for l_inf?**
>
> ---
>
> - Sorry for missing the details on the description about the type of attack method. We use the $L_\infty$-based FGSM to construct results in the main paper.  We denote the description for the type of attack to clarify in the main paper.
>
> - We also perform $L_2$-based FGSM to construct the correlation result and identify that the similar trend as $L_\infty$-based FGSM. In Table1, we provide the slope of the regression line, and P-value of the fitted line to verify the significance. Table 1 presents the PRS Ratio and robustness score are inversly correlated and shows all slopes are statistically significant (P-value $<$ 0.05).
>
> - We also provide slopes for other properties (e.g. relationship between the PRS ratio and sparsity) and corresponding P-values in Appendix E.
>
> **Table 1.** Slopes and P-value of each regression line between PRS ratio and $L_2$ based FGSM.
>
> |         |          |   CNN-6  |          |          |  VGG-16  |          |          | ResNet-18 |          |
> |:-------:|:--------:|:--------:|:--------:|:--------:|:--------:|:--------:|:--------:|:---------:|:--------:|
> |         |   MNIST  |  fMNIST  |  CIFAR10 |   MNIST  |  fMNIST  |  CIFAR10 |   MNIST  |   fMNIST  |  CIFAR10 |
> |  Slope  |    -0.18 |    -1.15 |    -0.48 |    -5.58 |    -5.80 |    -2.42 |    -2.92 |     -1.91 |    -0.75 |
> | P-value | 2.04E-03 | 1.34E-18 | 2.29E-15 | 2.79E-03 | 4.86E-12 | 6.53E-07 | 7.79E-04 |  1.27E-02 | 1.29E-18 |

---

> > ### Comment · Reviewer_5u8a · 2021-11-19
> > **Answer**
> >
> > Thanks for the detailed response.
> >
> > **Q1:**
> > I think the fact that the models differ in batch size should be made clear in the main paper not just the appendix.
> >
> > **Q2:**
> > Thanks for the explanation. Given this information about using different points in training as different models I would suggest changing table 3 in the appendix to list the results by batch size and episode instead of batch size and random seed. Then the different random seeds can be used to produce meaning full confidence intervals.
> >
> > **Q3:**
> > If $L_\infty$ is used then I find the choice of $\epsilon$ strange as the lowest value in figure 3 ($\epsilon = 0.05$) is already larger than what is normally used in the literature for Cifar-10 (e.g. $8/255 \approx 0.03$ is used on https://robustbench.github.io/) that together with using FGSM which is known to overstimate robustness [1] makes it hard to conclude much from the results.
> >
> > What $\epsilon$ value was used for the $L_2$ attacks in table 1?
> >
> >
> >
> > [1] Wang, Yisen, et al. "On the Convergence and Robustness of Adversarial Training." ICML. Vol. 1. 2019.

---

> ### Comment · Reviewer_5u8a · 2021-11-19
> **Update of review**
>
> I have increased my rating to 5. I think the authors have clarified a lot of statements and done more analysis. I still think there are a few outstanding issues stopping it from being above the acceptance threshold. Mainly
> * the confusing use of confidence intervals based on different times in training
> * using very high values of $\epsilon$ and FGSM as the methods to report robust accuracy in the main paper
> * no investigation into the advantage of using PRS instead of the robust accuracy directly.

---

> > ### Author Response · Authors · 2021-11-20
> > **Response to the update of review (2/2)**
> >
> > ---
> >
> > **Q4. No investigation into the advantage of using PRS instead of the robust accuracy directly.**
> >
> > ---
> >
> > - In our work, we categorize the successful attack samples into two groups (in test dataset): (1) samples included in the training PR, and (2) excluded from the training PR. In particular, we empirically show that the exclusion group is more vulnerable to the gradient-based adversarial attack (Section 5 in the main paper and Appendix I).
> >
> > - From these observations, we provide one method to increase the robustness without additional post-training. The main concept of the method is that if the sample is judged as the exclusion group, we trace which decision boundaries (DBs) prevent the sample to be included in the training PR. We note that the DBs are components to comprise the decision region (DR).
> >
> > - For example, let the indicator vector of training PRS $V{=} $\{$ [\textbf{+1},+1,+1] $}$ $ and the test sample $\bar{x}$ mapped into the specific DR with its indicator vector, $v_h{=}[\textbf{-1}, +1, +1]$. This test sample is not included in the train PR, because the test sample is located on the opposite side of first DB. Our simple idea, to modify this sample to be regarded as if the test sample is located in the train PR, is to reverse the sign of corresponding feature value by multiplying -1. We provide the pseudo code to clarify the method.
> >
> >
> > ---
> > > **Input**: training dataset $\mathbf{X}$, test sample $\bar{x}$ and target layer $l$
> > **Parameter**: the number of most similar vectors $K$
> > **Output**: modified output $y$
> >
> > 1. Obtain the feature embedding $h=f_{l:1}(\bar{x})$ for given $\bar{x}$
> > 2. Compute the boundary indicator $v_h=sign(h)$
> > 3. Compute a training set of the boundary indicator vector $V=${$sign(f_{l:1}(x))| \quad \forall x\in \mathbf{X}$}$ $
> > 4. Define $I = \emptyset$
> > 5. For $v$ in $K$ most similar vector to $v_h$ in $V$ with similarity metric $v^Tv_h$:
> > 6. $\quad$ $I = I \cup ${$ i|v_h^i \neq v^i, \quad \forall i \in [1,2,...,D_l] $}$ $
> > 7. Reverse embeddings $h^i=(-1)*h^i, \quad \forall i \in I$
> > 8. Compute modified output $y=f_{L:l+1}(h)$
> > 9. Return $y$
> >
> > ---
> >
> > - Table 1 shows the results for this reverse method on various architectures trained with CIFAR-10 (the 300th epoch). We can identify that this method can improve the robust accuracy maintaining the test accuracy without post-training. The confidence intervals are computed over 5 random seeds.
> >
> > - We think that these results can give the insight for the robustness from the perspective of DB and DR. We believe that the PRS concept can contribute to the improvement of robustness in neural networks.
> >
> > **Table 1.** Comparison of Test accuracy (Acc.) and robust accuracy between the clean model and the modified model ($K$=100)
> >
> > |   Model   |  Batch Size |  Clean Acc.   |  Modified Acc.    |  Clean Robust Acc. |  Modified Robust Acc. |
> > |:---------:|------------:|--------------:|--------------:|-------------------:|------------------:|
> > |   CNN-6   |         128 |  76.496±0.009 |  61.802±0.062 |       58.172±0.026 |  **74.680±0.068** |
> > |           |        2048 |  73.594±0.009 |  70.748±0.010 |       33.248±0.016 |  **41.778±0.018** |
> > | ResNet-18 |         128 |  77.314±0.002 |  77.036±0.003 |       37.792±0.025 |  **55.590±0.061** |
> > |           |        2048 |  72.216±0.009 |  71.892±0.009 |       28.684±0.012 |  **70.634±0.013** |
> > |   VGG-16  |         128 |  79.868±0.011 |  79.604±0.012 |       40.528±0.093 |  **51.704±0.154** |
> > |           |        2048 |  75.006±0.030 |  74.878±0.030 |       37.854±0.134 |  **45.058±0.175** |
> >
> > ---
> > Please kindly let us know if we address your concerns. We would love to try our best to address your concerns, so please feel free to let us know.
> > Thank you!

---

> > > ### Comment · Reviewer_5u8a · 2021-11-23
> > > **Thanks again**
> > >
> > > Thanks for the further explanation and changes. I've raised my rating to 6.

---

> > ### Author Response · Authors · 2021-11-20
> > **Response to the update of review (1/2)**
> >
> > Thank you for updating the score. Your feedback really helped us on improving our work. We are grateful for your insightful comments.
> >
> > ---
> >
> > **Q1. I think the fact that the models differ in batch size should be made clear in the main paper not just the appendix.**
> >
> > ---
> >
> > - Thanks for the comment. We updated the main paper to clarify the difference between the models (Network A and B).
> >
> > - We also modified the experimental setup in the main paper for better understanding.
> >
> > ---
> >
> > **Q2. Thanks for the explanation. Given this information about using different points in training as different models I would suggest changing table 3 in the appendix to list the results by batch size and episode instead of batch size and random seed. Then the different random seeds can be used to produce meaning full confidence intervals (the confusing use of confidence intervals based on different times in training).**
> >
> > ---
> >
> > -  We agree that changing table 3 in Appendix group by batch size and episode (epoch) helps readers to better understand. We updated Table 3 in Appendix to list the results by batch size and episode (epoch).
> >
> > ---
> >
> > **Q3: If L_inf is used then I find the choice of  strange as the lowest value in figure 3 () is already larger than what is normally used in the literature for Cifar-10 (e.g.  is used on https://robustbench.github.io/) that together with using FGSM which is known to overstimate robustness [1] makes it hard to conclude much from the results. What epsilon value was used for the L_2 attacks in table 1 ? (using very high values of and FGSM as the methods to report robust accuracy in the main paper)**
> >
> > ---
> >
> > - We agree that there is a possibility to overestimate the robustness with the value of $\epsilon$ that we had used, although we intended to show the difference in robustness with large perturbations. We attached all experimental results related to the robust accuracy in the main paper with updated $\epsilon$ used in the robust bench to validate the observed relationship holds.
> >
> > - We changed the used attack method in the main paper from FGSM to PGD ($L_\infty$) attack. The Figure 4, 5, 6, and 9 are also changed based on the PGD attack.
> >
> > - For the $L_2$-based FGSM attack, we previously used various $\epsilon$ ([0, 0.7, 1.3, 2.6, 4.0]) for the curve of robust accuracy. The value of AUC of each curve is used to calculate the robustness score.
> >
> > - With concerns about large $\epsilon$, we perform $L_2$-based FGSM attack with $\epsilon=0.5$ over CIFAR-10 (random seed 375) which is used in the robust bench. In Table1, we provide the slope of the regression line, and P-value of the fitted line to verify the significance.
> >
> >
> > **Table 1.** The slope of regression line and P-value for the significance test.
> >
> > |         |          |   CNN-6  |          |          |  VGG-16  |          |          | ResNet-18 |          |
> > |:-------:|:--------:|:--------:|:--------:|:--------:|:--------:|:--------:|:--------:|:---------:|:--------:|
> > |         |   MNIST  |  fMNIST  |  CIFAR10 |   MNIST  |  fMNIST  |  CIFAR10 |   MNIST  |   fMNIST  |  CIFAR10 |
> > |  Slope  |   0.01   |   -0.11  |   0.00   |   -0.17  |   -0.35  |   -0.11  |   -0.10  |   -0.26   |   -0.13  |
> > | P-value | 3.95E-02 | 2.18E-11 | 5.84E-01 | 7.44E-02 | 9.27E-06 | 2.31E-01 | 7.80E-07 |  2.36E-04 | 3.86E-18 |

---

### Official Review · Reviewer_Yvre · 2021-11-01

**Correctness:** 3
**Technical Novelty And Significance:** 2
**Empirical Novelty And Significance:** 2
**Recommendation:** 6
**Confidence:** 2

**Main Review:**

The paper is well-written and clear. The newly introduced geometry metric, the so-called Populated Region Set (PRS) ratio seems promising and intuitive. Although I am not knowledgeable of the related work on the interaction between decision boundaries and model's robustness, I  find the type of analysis/comparisons made (which I indicated in the summary) insightful.

The conclusions that are drawn related to the decision regions and based on the empirical evidence are consistent with what one may indeed intuit. As useful as I find its intellectual contribution, I am not fully convinced on the informativeness of the PRS ratio when it comes to more challenging settings. For instance, what happens when the number of classes is large? In the experiments, the datasets in consideration have only 10 classes each, which is limiting. As an immediate thought, I would expect that the PRS ratio is generally high in cases where the label space is large. What do the authors think about it?


_Typos:_ "We verify that the the size ...", "Traini accuracy" in Figure 2 first column


**Summary Of The Paper:**

This paper aims to understand the robustness of DNNs from the perspective of decision regions. Towards that, the authors introduce a new metric, the so-called Populated Region Set (PRS) whose ratio is later used to investigate the robustness of a selection of DNNs empirically. Based on the respective empirical evidence, the paper claims that the lower PRS ratio (roughly #decision regions with at least one training data point/size of training data) leads to enhanced robustness, better representation of test instances in the populated regions, and learns the sparse feature representation. The empirical evidence is collected using the models CNN, ResNet-18, and VGG-16 over datasets MNIST, F-MNIST, and CIFAR-10 and under perturbations with various $\epsilon$ and untargeted/targeted attacks.

**Summary Of The Review:**

The analysis of the interaction between the robustness of ML models and the Populated Region Set (PRS) ratio seems promising in understanding the geometrical interpretation of robustness. Although having an extension to the current experiments would make the use of such a metric more convincing, I am slightly leaning towards acceptance.

---

> ### Author Response · Authors · 2021-11-16
> **Response to Reviewer Yvre**
>
> We sincerely appreciate your time and efforts in reviewing our paper, as well as the constructive comments. We have revised the manuscript by faithfully reflecting your comments. We respond to your comments below.
>
> ---
>
> **Q1. I am not fully convinced on the informativeness of the PRS ratio when it comes to more challenging settings. For instance, what happens when the number of classes is large? In the experiments, the datasets in consideration have only 10 classes each, which is limiting. As an immediate thought, I would expect that the PRS ratio is generally high in cases where the label space is large. What do the authors think about it?**
>
> ---
>
> - We also agree that the PRS ratio on the penultimate layer will increase if the dataset with larger number of class but same size of samples is used to train the model (i.e., The number of samples in each class decreases). To verify our hypothesis, we design a small experiment with CIFAR-10 dataset. First, we prepare the subsampled CIFAR-10 dataset with $n$ classes, by sampling 10k/$n$ number of samples for each class.
>
> - Then we train the CNN-6 (Adam optimizer with learning rate of 1e-3, batch size 128, 300 epochs, and 5 random seeds) and measure the average of PRS ratio for various $n$. Table 1 presents the PRS ratio increases as the number of classes increases. We conjecture that this phenomenon may caused by the number of classes, because the model seems to be needed more complex feature representation to classify more various classes.
>
> **Table 1.** The PRS ratio for the number of class.
>
> | # of class    |         2        |         5        | 7               | 10              |
> |---------------|------------------|------------------|-----------------|-----------------|
> | PRS Ratio     | 0.051$\pm$0.044  | 0.698$\pm$0.115  | 0.979$\pm$0.014 | 0.996$\pm$0.003 |
> | Test accuracy | 0.8258$\pm$0.002 | 0.7816$\pm$0.002 | 0.751$\pm$0.001  | 0.616$\pm$0.001 |
>
>
> ---
>
> We sincerely thank you for the insightful comments since it helped us to further improve the discussions in the paper.
> Please let us know if there is anything else we should address, or misunderstood.
>
> ---

---

### Official Review · Reviewer_pYmY · 2021-11-02

**Correctness:** 3
**Technical Novelty And Significance:** 3
**Empirical Novelty And Significance:** 3
**Recommendation:** 5
**Confidence:** 4

**Main Review:**

##########################################################################

Pros:

* Characterizing decision boundaries is an interesting topic. I think there is a lot of interesting empirical work to be done in this area.

* Figure 2 is interesting: the PRS ratio clearly decays to zero for network B.

##########################################################################

Cons:

* Why no PGD attacks in the robustness studies?

* Relationship is pretty weak for CNN-6 in Figure 4. Any thoughts why?

* In Section 6, the authors use the word "intensive" to describe t-SNE plots. What do the authors mean by this? For instance the authors write: "In Figure 6, we can identify that the model with the low PRS ratio has more intensive feature representation for each class." Is there a quantitative measure for this?


* Regression slopes are only visualized, not reported quantitatively. I would also like the see significance tests.

##########################################################################

Questions during rebuttal period:


* "Robustness score", the area under the robustness curves, is okay to report, but to the best of my knowledge it is non-standard in the robustness community. Epsilon sweeps are included in the Supplementary.

How is it actually possible that you can compute your proposed measure? Why does the "curse of dimensionality" not apply?

* What is the relationship of your work and those which analyze the linear regions of decision surfaces? E.g Montufar et al. (NeuIPS 2014)
https://arxiv.org/abs/1402.1869


##########################################################################

Post rebuttal period: I thank the authors for their thorough response. Having their response and the other reviews, I still harbor some doubts about the validity and usefulness of the proposed PRS metric and its empirical evaluation. I have decided to keep my score marginally below acceptance.

**Summary Of The Paper:**

This work empirically studies for deep networks the relationship between (1) model robustness and (2) the decision surface. A novel metric is proposed, the Populated Region Set (PRS) metric, essentially the number of regions in decision space which have at least one training sample. The authors claim the metric has a "strong relationship" to robustness, as measured by correlation, and present a number of experiments to support their claim.

**Summary Of The Review:**

Interesting empirical work but needs improvement and clarification.

---

> ### Author Response · Authors · 2021-11-16
> **Response to Reviewer pYmY (4/4)**
>
> ---
>
> **Q7. What is the relationship of your work and those which analyze the linear regions of decision surfaces? E.g Montufar et al. (NeurIPS 2014)
> " On the number of linear regions ~" (https://arxiv.org/abs/1402.1869)**
>
> ---
> - Montufar et al. provide the mathmatical framework to quantify the expressive power of the deep neural network. Especially, the proposed framework represents the expressivity in the perspective of the maximal number of the linear regions decided by the depth and width of the architecture of the model. The direction of analysis which uses the decision region of network is simliar as our work to understand the characteristics of deep neural network.
>
> - The difference between previous work and ours is that our work focuses on the practical decision region which the trained network actually utilizes. We observe that networks can use different number of decision regions although they have the same number of maximal regions. To analyze this phenonmenon, we also devise the novel concept *Populated Decsion Region Set (PRS)*. Especially, we provide extensive experiments to investigate the relationship between PRS ratio and various properties of DNNs.
> ---
>
> We sincerely thank you for the insightful comments since it helped us to further improve the discussions in the paper.
> Please let us know if there is anything else we should address, or misunderstood.
>
> ---
>
> **Reference**
>
> [1] Shaeiri, Amirreza et al. “Towards Deep Learning Models Resistant to Large Perturbations.”, ICLR 2018
> [2] Croce, Francesco et al. “Reliable evaluation of adversarial robustness with an ensemble of diverse parameter-free attacks.”, ICML 2020

---

> ### Author Response · Authors · 2021-11-16
> **Response to Reviewer pYmY (3/4)**
>
> ---
>
> **Q4. Regression slopes are only visualized, not reported quantitatively. I would also like the see significance tests.**
>
> ---
>
> - Thank you for the insightful comment. We provide the quantitative results for the calculated regression lines (in Figure 4, 8, 10, and 12 of the main paper) which represent the relationship between the PRS ratio and each property. We provide (1) slope of the regression line, and (2) P-value of the fitted line to verify the significance. Table 1 presents the result of slope in each case. The inverse correlation is observed in most investigated properties. The relationship between the PRS ratio and the sparsity on ResNet-18 shows independence in all dataset. We conjecture that skip connection or batch normalization can cause this phenomenon. Table 2 indicates the result of P-value (computed by *statsmodel* package) from the significance test of the slope of the regression line. We can identify that the most slopes are statistically significant (P-value $<$ 0.05).
>
>
>
> **Table 1.** Slopes of the each regression line between the PRS ratio and properties. FGSM indicates the robust score based on FGSM defined in the main paper. PGD indicates the robust accuracy based on PGD attack with fixed $\epsilon$ (MNIST=0.3, F-MNIST=0.1, and CIFAR10=0.0313). IR/CS indicates an inclusion ratio and cosine similarity respectively.
>
> |    Model   |   Dataset  | FGSM/PRS | PGD/PRS | Sparsity/PRS | IR/PRS | CS/PRS |
> |:----------:|:----------:|:--------:|:-------:|:------------:|:------:|:------:|
> |            |    MNIST   |    -0.72 |    -0.3 |        -0.49 |  -0.72 |  -0.76 |
> |   CNN-6    |   F-MNIST  |    -1.44 |   -0.45 |        -0.23 |  -0.79 |  -0.58 |
> |            |  CIFAR-10  |    -1.05 |   -0.23 |        -0.15 |   -0.8 |  -0.65 |
> |            |    MNIST   |    -20.4 |  -14.32 |        -1.99 |  -0.53 |  -14.7 |
> |   VGG-16   |   F-MNIST  |    -8.65 |   -2.52 |        -2.24 |  -0.48 |  -3.38 |
> |            |  CIFAR-10  |    -4.11 |   -0.86 |         -0.9 |  -0.79 |   0.28 |
> |            |    MNIST   |    -4.61 |    -0.8 |         0.05 |  -0.53 |  -2.88 |
> | ResNet-18  |   F-MNIST  |    -2.75 |    -0.9 |         0.08 |  -0.65 |  -2.24 |
> |            |  CIFAR-10  |    -0.57 |   -0.11 |        -0.03 |  -0.71 |  -1.29 |
>
>
> **Table 2.** P-value from the significance tests for each regression line.
>
> |    Model   |   Dataset  | FGSM/PRS |  PGD/PRS | Sparsity/PRS |   IR/PRS  |  CS/PRS  |
> |:----------:|:----------:|:--------:|:--------:|:------------:|:---------:|:--------:|
> |            |    MNIST   | 3.92E-06 | 9.43E-05 | 3.87E-48     | 1.09E-126 | 5.17E-29 |
> |   CNN-6    |   F-MNIST  | 4.42E-22 | 1.28E-44 | 9.84E-13     | 1.78E-133 | 2.43E-46 |
> |            |  CIFAR-10  | 3.57E-23 | 2.36E-22 | 1.92E-11     | 5.45E-136 | 4.35E-54 |
> |            |    MNIST   | 1.53E-03 | 7.01E-05 | 2.91E-06     | 4.66E-43  | 2.70E-08 |
> |   VGG-16   |   F-MNIST  | 1.93E-13 | 4.38E-18 | 3.79E-43     | 2.22E-24  | 4.11E-19 |
> |            |  CIFAR-10  | 6.03E-12 | 2.43E-07 | 9.23E-12     | 2.07E-09  | 2.92E-01 |
> |            |    MNIST   | 5.40E-03 | 1.38E-02 | 4.16E-01     | 1.58E-98  | 1.36E-11 |
> | ResNet-18  |   F-MNIST  | 6.10E-03 | 3.25E-05 | 1.96E-02     | 6.38E-38  | 1.66E-16 |
> |            |  CIFAR-10  | 2.17E-07 | 5.23E-13 | 2.87E-03     | 2.87E-35  | 4.20E-14 |
>
>
> ---
>
> **Q5. "Robustness score", the area under the robustness curves, is okay to report, but to the best of my knowledge it is non-standard in the robustness community. Epsilon sweeps are included in the Supplementary.**
>
> ---
>
> - Thank you for the constructive comments. To follow the rule for robustness community and improve the clarification, we provide the graphs of epsilon sweep on the FGSM for robust accuracy in Appendix J - R.
>
> ---
> **Q6. How is it actually possible that you can compute your proposed measure? Why does the "curse of dimensionality" not apply?**
>
> ---
>
> - To compute $PRS(\mathbf{X}, f, l)$ in the Definition 3, we need to collect the sign of feature vector for dataset $\mathbf{X}$ at the target layer $l$ first.
> For each data $x\in\mathbf{X}$, we feed $x$ to the model $f$ to get the sign of feature vector at the target layer index $l$.
> Then we take the unique over the collected sign vectors, to get the populated region set, $PRS(\mathbf{X},f,l)$.
>
> - If we need to verify the area of the decision region, it requires exponentially many samplings, which is related to the curse of dimensionality.
> However, our point of interest is to collect the decision regions which are actually populated by given dataset, and its computaional complexity is linearly proportional to the size of dataset. As a result, we think that our proposed measure has weak relationship with the curse of dimensionality.

---

> ### Author Response · Authors · 2021-11-16
> **Response to Reviewer pYmY (2/4)**
>
> ---
>
> **Q3. In Section 6, the authors use the word "intensive" to describe t-SNE plots. What do the authors mean by this? For instance the authors write: "In Figure 6, we can identify that the model with the low PRS ratio has more intensive feature representation for each class." Is there a quantitative measure for this?**
>
> ---
> - We agree that the inappropriate term "intensive" causes an ambiguity of the contents, so modified the description to clarify the contents.
> Our intention was to represent that a network with a low PRS ratio tends to show more densely located samples per each class in the feature space on the penultimate layer. We provided the t-SNE figures as an method to visualize this property.
>
> - To support our hypothesis, we also provide the quantitative experiment to measure the denseness of the feature representations in each class over the networks with different PRS ratio. In this experiment, we measure the ratio of same class samples in the feature space based on $K$ nearest neighbors. Let the training dataset $\mathbf{X}$ and data pair $(x_i,y_i)\in \mathbf{X}$. The denseness for the given data pair $(x,y)$ is defined as,
>
> $$D(\mathbf{X},K)=\frac{1}{|\mathbf{X}|}\sum_{x,y \in \mathbf{X}}\frac{1}{K}\sum_{i\in NN_K(x, \mathbf{X})} \mathbf{I}(y_i=y), \quad $$
>
> - where $NN_K(x,\mathbf{X})$ returns indices of $K$ nearest neighbors for feature representation of the given input $f_{L-1:1}(x)$ from training dataset $\mathbf{X}$ and $\mathbf{I}(\cdot)$ is the indicator function.
> Table 1 presents the average denseness for various random seeds mentioned in Table 2 of Appendix A with various $K$. The experiment is performed in CNN-6 with low/high PRS ratio (we use the trained models with batch size of 128 and 2048 to control the PRS ratio).
> We identify that the low PRS ratio cases (a1, b1, c1) have higher densness compared to the high PRS ratio cases (a2, b2, c2) in almost $K$. For MNIST dataset, it is difficult to discriminate the difference of densness along the magnitude of PRS ratio.
>
>
> **Table 1.** Average of the denseness per each class in feature space for CNN-6 with low PRS ratio and high PRS ratio on three datasets, (a) MNIST, (b) F-MNIST and (c) CIFAR-10. Odd rows (a1, b1, c1) indicate the model with low PRS ratio and even rows (a2, b2, c2) indicate the model with high PRS ratio.
>
> |      |  PRS Ratio  |      10     |      50     |     100     |     200     |     300     |
> |:----:|:-----------:|:-----------:|:-----------:|:-----------:|:-----------:|:-----------:|
> |(a1) | 0.035${\pm}$0.029 | **0.999${\pm}$0.001** | **0.998${\pm}$0.001** | 0.997${\pm}$0.001 | 0.996${\pm}$0.002 | 0.995${\pm}$0.002 |
> |(a2) | 0.669${\pm}$0.088 | 0.999${\pm}$0.001 | **0.998${\pm}$0.001** | **0.998${\pm}$0.001** | **0.997${\pm}$0.002** | **0.996${\pm}$0.002** |
> |(b1) | 0.022${\pm}$0.007 | **0.981${\pm}$0.006** | **0.972${\pm}$0.008** | **0.966${\pm}$0.009** | **0.957${\pm}$0.010** | **0.949${\pm}$0.011** |
> |(b2) |0.946${\pm}$0.018 | 0.969${\pm}$0.003 | 0.955${\pm}$0.003 | 0.948${\pm}$0.004 | 0.938${\pm}$0.004 | 0.932${\pm}$0.004 |\hline
> |(c1)  | 0.006${\pm}$0.000 | **0.945${\pm}$0.006** | **0.920${\pm}$0.007** | **0.903${\pm}$0.007** | **0.877${\pm}$0.007** | **0.857${\pm}$0.007** |
> |(c2) | 1.000${\pm}$0.000 | 0.939${\pm}$0.007 | 0.904${\pm}$0.008 | 0.883${\pm}$0.009 | 0.855${\pm}$0.010 | 0.836${\pm}$0.011|

---

> ### Author Response · Authors · 2021-11-16
> **Response to Reviewer pYmY (1/4)**
>
> We sincerely appreciate your time and efforts in reviewing our paper, as well as the constructive comments. We respond to each of your comments one by one.
> You can check the revised Figures mentioned in answer from the updated main paper and Appendix.
>
>
> ---
>
> **Q1. Why no PGD attacks in the robustness studies?**
>
> ---
>
> - At first, we had considered FGSM for computational efficiency.
>
> - To validate the observed relationship between the PRS ratio and the robustness also holds for the PGD [1] attack, we perform the same experiment. After the adversarial attacks, we measure the slope of regression line and perform significance test which you mentioned in Q2. Table 1 indicates the result of experiment. We identify that in all architectures and datasets, the inverse correlation is also valid for PGD attack. The scatter plots and their regression lines are attached in Figure 4 of main paper. We also updated the main attack from FGSM to PGD method for Fig 3,4,5,6, and 9.
>
> **Table 1.**  The slope of regression line and P-value for significance test under PGD attack.
>
> |   Model  | Dataset |  Slope  |  P-value |
> |:--------:|:-------:|:-------:|:--------:|
> | 　       | MNIST   | -0.30   | 9.43E-05 |
> | CNN-6   | fMNIST  | -0.45   | 1.28E-44 |
> | 　       | CIFAR10 | -0.23   | 2.36E-22 |
> | 　       | MNIST   | -14.32  | 7.01E-05 |
> | VGG-16    | fMNIST  | -2.52   | 4.38E-18 |
> | 　       | CIFAR10 | -0.86   | 2.43E-07 |
> | 　       | MNIST   | -0.80   | 1.38E-02 |
> | ResNet-18 | fMNIST  | -0.90   | 3.25E-05 |
> | 　       | CIFAR10 | -0.11   | 5.23E-13 |
>
> - We also measure the robust accuracy for various attacks (PGD and Auto attack (AA) [2]) in two networks with Low/High PRS ratio. For the experiments, we adopt 20-step and take the magnitude of $\epsilon$ as follow: MNIST=0.3, F-MNIST=0.1, and CIFAR10=0.0313 on $L_\infty$-based attack. The used models are selected at the 200th epoch with random seed 907. Table 2 presents the robust accuracy under the adversarial attacks.
>
> - We validate that observed relationship between PRS ratio and robustness holds for the various attacks in almost cases and provide this result in Table 7 of Appendix G.
>
>
> **Table 2.** Robust accuracy for various attack methods.
>
> |   Model   | Dataset | Batch Size | PRS Ratio |  Clean |  FGSM  |   PGD  |   AA   |
> |:---------:|:-------:|:----------:|:---------:|:------:|:------:|:------:|:------:|
> |           |  MNIST  |     128    |   0.201   | 99.46  | **83.54**  | **39.79**  |  0.00  |
> |           |         |    2048    |   0.679   | 99.39  | 37.11  | 11.81  |  0.00  |
> |   CNN-6  |  fMNIST |     128    |   0.195   | 92.24  | **52.63**  | **34.05**  |  0.00  |
> |           |         |    2048    |   0.977   | 91.05  | 28.39  |  2.67  |  0.00  |
> |           | CIFAR10 |     128    |   0.022   | 77.60  | **44.71**  | **41.17**  | **23.57**  |
> |           |         |    2048    |   1.000   | 75.66  | 40.10  | 25.92  |  7.48  |
> |           |  MNIST  |     128    |   0.001   | 99.18  | 54.22  | 36.36  |  0.00  |
> |           |         |    2048    |   0.024   | 99.35  | **55.59**  | **36.68**  |  0.00  |
> |   VGG-16  |  fMNIST |     128    |   0.012   | 92.66  | **45.09**  | **30.60**  | **14.28**  |
> |           |         |    2048    |   0.060   | 92.44  | 41.31  | 17.14  |  7.99  |
> |           | CIFAR10 |     128    |   0.012   | 77.90  | **44.40**  | **33.68**  | **21.08**  |
> |           |         |    2048    |   0.106   | 75.05  | 39.92  | 30.03  | 14.22  |
> |           |  MNIST  |     128    |   0.013   | 99.34  | **51.15**  | **24.88**  |  0.00  |
> |           |         |    2048    |   0.078   | 99.37  |  0.40  |  0.00  |  0.00  |
> | ResNet-18 |  fMNIST |     128    |   0.027   | 90.75  | **31.56**  | **19.94**  |  **0.60**  |
> |           |         |    2048    |   0.078   | 90.29  | 26.55  | 12.98  |  0.00  |
> |           | CIFAR10 |     128    |   0.073   | 75.40  | **40.27**  | **29.87**  | **28.52**  |
> |           |         |    2048    |   0.334   | 72.08  | 32.61  | 24.51  | 23.45  |
>
>
> ---
>
> **Q2. Relationship is pretty weak for CNN-6 in Figure 4. Any thoughts why?**
>
> ---
>
> - We also observed the weak inverse correlations in the FGSM and the PGD attack cases, and we think that the architecture of CNN-6 can be one of the reasons. The architecture of CNN-6 is relatively shallow and simple compared to other architectures (e.g., VGG-16 or ResNet-18).

---

### Decision · Program_Chairs · 2022-01-20

**Decision:**

Reject

**Comment:**

This work proposes a concept called Populated Region Set (PRS) as a measure of robustness of deep neural networks (DNNs). The paper provides a suit of empirical results to demonstrate the strong correlation between the PRS ratio and adversarial robustness of DNNs. The authors made great efforts on addressing reviewers' concern, which is greatly appreciated. However, the theory of the work is a bit thin, and it leaves a number of outstanding issues unaddressed. For example, it is not clear the practical advantage of calculating PRS over the direct measure of robust accuracy. What new and better computational procedure can be constructed based on PRS? We encourage the authors keep improving their work for future submission.